# The Recipe for Success: Repin's Painting *Barge-Haulers on the Volga* and Stasov's Conception of Russian Art

**Ariela Shimshon**

The Department of the Arts, Faculty of Humanities and Social Sciences, Ben-Gurion University of the Negev, P.O. Box 653, Beer-Sheva 8410501, Israel; arielas@bgu.ac.il

**Abstract:** *Barge-Haulers on the Volga* is one of the most famous works of the Russian realist painter Ilya Repin. As I demonstrate in this article, on the one hand, it brought Repin resounding success and, on the other, it molded his creative conception. The Russian art critic Vladimir Stasov outlined the artist's success. In March 1873, Stasov's poetic depiction of Repin's painting, where he expressed his admiration for Repin's talent, focusing on specific aspects that he contended had to be included in a perfect work of Russian art, was published in the daily newspaper *Sankt-Peterburgskie Vedomosti*. I attempt to show that Stasov's praise had a devastating effect on Repin's creative process. By examining Repin's post *Barge-Haulers* successful works for this pattern, I show how the painter tried to incorporate every one of the "ingredients" that Stasov outlined and ultimately created a typified group of paintings documenting life on the periphery of the Russian Empire with those features, which marked his entire career.

**Keywords:** Ilya Repin; Russian art; realism; *Barge-Haulers*; art critique; pictorial trademark; success; Vladimir Stasov





## 1. Introduction

*Barge-Haulers on the Volga* or *Burlaki na Volge* in Russian (Figure 1) is one of the most famous works by the Russian painter Ilya Repin, who later became the most iconic Russian realistic painter of his age. The painting, which depicts a group of eleven men dragging a barge, was shown at the Vienna World's Fair in 1873. It was due to the combination of the choice of subject (a simple mundane task) and its unusual dimensions (3 m wide) that it was chosen to represent Russian art in that major international exhibition. The painting had been purchased by Grand Duke Vladimir Alexandrovich in 1870, and in 1871 it won first prize in a competition of the Society for the Support of Artists held in St. Petersburg. During March 1873, before being sent to Vienna, the painting was exhibited in the annual exhibition of the Academy of the Arts in St. Petersburg (Valkenier 1990, p. 40), which is where Vladimir Stasov, the unquestioned authority on Russian art, saw it and was impressed. He subsequently expressed his admiration for the painting in an untitled article published in the letters column of the daily newspaper *Sankt-Peterburgskie Vedomosti* (Stasov 1873, p. 2).

At the time of the 1873 Vienna exhibition, Repin was a student at the Russian Im-perial Academy of Arts searching for his artistic style. As I demonstrate, The Barge-Haulers on the Volga brought Repin resounding success, becaming his "pictorial trademark," and molded his creative conception.

For Stasov, *The Barge-Haulers* was a visual expression of his own ideological expecta-tions. In his multilayered and lyrical description, Stasov focused on specific aspects that he claimed were essential components of a perfect work of Russian art. It was, in fact, in this early publication, that he outlined artistic features that Repin would later repeat time and again throughout his entire career. This was the beginning of a longstanding relationship between the painter and the critic, as one can discern from the correspondence between them, which lasted more than twenty years (1871–1894).

In a monograph about Repin, first published in 1937, Igor Grabar the art historian and painter, who was one of Repin's students, explained Stasov's involvement in his master's career. Grabar described the relationship between the two as a struggle between a painter of genius, who wanted to preserve the purity of his work, and the uncompromising art critic searching for art that would serve the national ideology. In Grabar's view, Kramskoi helped Repin develop his independent thinking, while Stasov restricted his outlook and assumed the role of apologist when Repin's art "clung" to his ideology. Stasov accepted only art in which the inherent idea was close to his personal convictions regarding the national agenda and these frameworks were too tight (Grabar 1964, pp. 234–42). Other Soviet art historians also acknowledged Stasov's influence on the Repin. Ilya Zil'bershteyn, for example, noted that Stasov led Repin to choose the genre of ideological realism (Zil'bershteyn 1945, p. 65). Olga Lyaskovskya described many of Stasov's reactions to Repin's various works from the very beginning of his career (Lyaskovskaya 1953, n.p.). She argued that with *Burlaki*, Stasov's appreciation of Repin intensified and that his critique of the painting symbolized a new era in Russian art. She contended that Repin was much less influenced by other people's opinions than is commonly thought. Elizabeth Valkenier notes two important mentors in Repin's life—Kramskoi and Stasov—but insists that Repin absorbed so many ideological influences in the 1860s that it is not easy to organize them into a single system (Valkenier 1990, p. 33). Despite the consensus among experts on Repin regarding Stasov's role in creating his aesthetic theory, the significance of Stasov's mentoring has not yet been thoroughly explored.

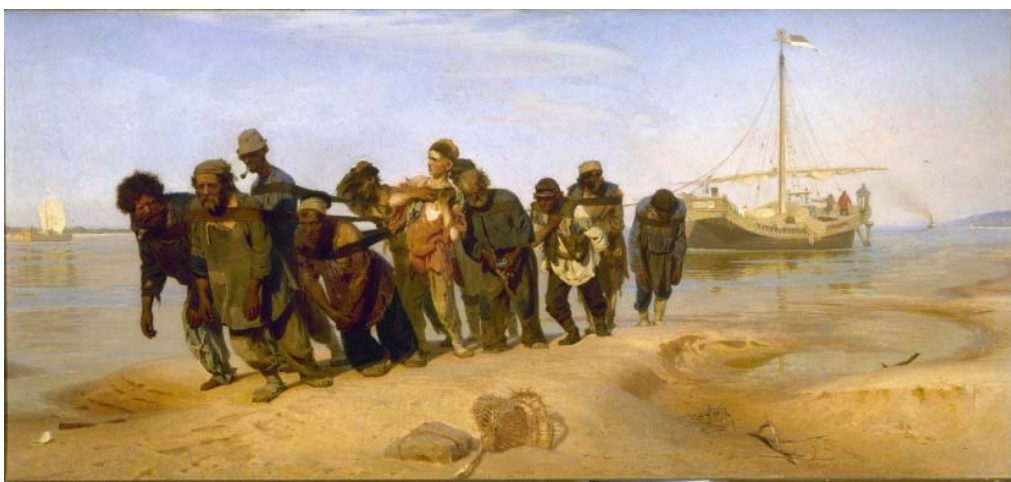

**Figure 1.** Repin, Barge Haulers on the Volga, 1870–1873, Saint Petersburg, The State Russian Museum. Digital Image: Wikimedia Commons, Public Domain.

Stasov's publication can be compared to a recipe in which one can find the most perfect ingredients available for a talented cook in order to prepare the perfect dish. In this essay, I discuss the extent to which Stasov's praise influenced Repin's creative process. I attempt to show that with his endless attempts to follow Stasov's guidelines, Repin adopted what might be called Stasov's "recipe," and that by doing so, constrained his artistic perception to a comfortable pattern that repeated itself throughout his entire career. By exploring Repin's post-*Barge-Haulers* successful works, which followed this pattern and received enthusiastic notices in the press, I show how the painter tried to adhere to every single 'ingredient' that Stasov mentioned to in his article and ultimately created a typified group of paintings portraying the lives of common people on the periphery of the Russian Empire. Those famous works, like *Barge-Haulers on the Volga*, epitomize Russian realist painting and marked Repin's entire career. In his review of the exhibition, Stasov focused on Repin and *Burlaki*. The work we know today as *Barge-Haulers on the Volga* is the second version of the scene. Grand Duke Vladimir Alexandrovich, then vice-president of the Academy of the Arts, was impressed by Repin's sketches of the Volga and commissioned the work,

granting the artist a stipend. Repin, who was convinced that the subject of the work was provocative and unsuitable for a member of the royal family, tried to sell it to someone else. He approached the collector Pavel Tretyakov, but the latter turned it down on the grounds that everybody knew that the work had been commissioned by the Grand Duke (Repin 1946, pp. 18–20; Grabar 1964, p. 234).

In *Burlaki* we see a group of eleven male figures, one of them just a boy. Their clothes are shabby and torn, they are dirty, with long, unkempt hair and beards, and they appear to be exhausted. Each of them has a wide strap round his chest with which he does his part in hauling a barge, shown to the right in the picture. The men are depicted making their way alongside shallow pools on a sandy riverside beach on a sunny day under a blue sky. The boy has a different look than the others—his figure is brightly colored, his head is raised, and he trying to pull off the strap, symbolizing youthful rebellion against the mindless surrender of the rest.

Stasov, who described the final of version of the painting in his article, had two objectives in mind: the first was to specify guidelines for Russian national art and the second was to declare that Repin had succeeded in realizing the potential of Russian national art. To achieve these objectives he described Repin's work at great length, declaring it the best work produced by a Russian painter to date, and "crowned" Repin as the ultimate Russian national painter: "According to the plan and expression of his painting, Mr. Repin is a significant, powerful artist and thinker, but at the same time he owns the expertise of his art with such strength, beauty and perfection, as hardly any other Russian artist before" (Stasov 1873, p. 2). Stasov praised Repin's courage for abandoning the attempt to create idealized beauty following the academic tradition. In effect, Stasov "recruited" Repin to his ideological conflict with the Academy of the Arts. Stasov constantly criticized what he considered the outdated management of the Academy, denouncing its choice of subjects for graduate projects from Greek mythology, which, he said, were irrelevant in the Russian reality and deriding the Academy's teaching methods.[1] He was a committed supporter of the Peredvizhniki, a movement of painters who had chosen to leave the Academy and ignored its exhibitions, which was established following the "the Revolt of the Fourteen" in 1863, when fourteen outstanding students refused to participate in the Academy's Gold Medal Competition.

At the same time, Stasov compared Repin to Nikolai Gogol, an author of the Russian Natural School, maintaining that they both expressed a nationalistic worldview in their depictions of reality. Gogol, who wrote about life on the periphery of the Russian Empire, came into his own in descriptions of food and gastronomic culture.[2] His stories feature detailed portrayals of traditional Russian dishes, stressing not only the taste of the food but also its aroma, how it was served, and how the Russians used to sit together at table—all things passed down from one generation to the next. Just as Gogol's depictions evoked a strong desire to participate in a traditional Russian meal, so too the wealth of description that characterizes Stasov's art criticism was designed to move the reader to "step inside" the work and connect to the characters, who led simple but authentic lives (Baluev 2010, pp. 138–40).

In writing about Repin's painting, Stasov made brilliant use of ekphrasis: his description of sounds like curses, of the young boy spitting, and of the shining rays of the sun convey the sensation of oppressive heat experienced by the haulers. He described the movement of the toiling men and their power in the joint effort. He praised the group's physical strength, comparing the barge haulers to oxen and to Hercules: "What glances from untamed eyes, what distended nostrils, what iron muscles . . . " Stasov devised a background for each of the figures, describing their reactions to the events taking place around them: "Behind them is a third epic hero . . . this man, it seems, has been everywhere, trying his luck and experiencing life in all corners of the earth, and now himself has begun to look like an Indian or Ethiopian of sorts." He allotted a pivotal role to the young man positioned in the center, whom he saw as a symbol of youthful rebellion against the previous generation's submissiveness.

The ingredients required for the recipe for the perfect national artwork can be "distilled" from Stasov's article. The first one that he related to was the narrow landscape, serving only as a backdrop. This was important in order to distance national art from the academic art traditions: paintings created in the Academy of the Arts' traditional style featured glamorized ideal settings, overloaded with details that, he wrote, were alien to the Russian viewer.The second ingredient was the human group, made up of various powerful figures. For Stasov this feature served to stress the power inherent in a united group, especially one that consisted of people with individual strengths and fascinating personal stories, who together comprise the Russian national identity. The third ingredient was movement—the essence of the group's strength in joint activity. The fourth was a central figure, the work's protagonist, which enabled Stasov to point the viewer toward an aspiration for a better future. The last ingredient was the realistic depiction of the figures. Stasov delineated the central and contemporary subject of the national artwork as "life in the remote and unknown Russia," whose essence was alive in Repin. He accepted the suggestion of poverty in Repin's work but gave it a novel interpretation: "Repin created his work to display the life of the real Russia, not to stir pity".

## 2. Stasov's Ideology and Mentoring

Stasov's enthusiastic approach to Repin's work can be accounted for by an understanding of the character of the art critic and his commitment to a positive Russian national ideology. For Stasov the ideological message as a tool for expressing the expectations of the Russian people was primary and he criticized artists who disagreed with him in the newspaper and in letters. This was also true of his association with Repin.

The relationship between Stasov and Repin began in 1871 when the latter was a student at the Academy and Stasov was working in the art department of the Imperial Public Library (Karenin 1927, p. 227). Stasov gave Repin advice and helped him find material for his works (Repin 1944, p. 204). Although they remained friends until Stasov's death in 1906, they did sometimes disagree. One serious ideological quarrel arose in 1894, when Stasov broke off his relations with Repin owing to the painter's acceptance of an invitation to teach at the Academy of Arts, Stasov's *bête noire*. Stasov, who was born into an aristocratic Russian family, was incapable of understanding that for Repin, the son of a military settler who had arrived in Moscow with a hundred rubles in his pocket, a teaching position at the leading art institution was a testimony to his success. In letters to Stasov after they became reconciled, Repin wrote that he understood that ideological principles were more important to Stasov than personal friendships (Repin's letter of 13 June 1894, in Repin and Stasov 1949, pp. 205–9).

Stasov saw himself as an ideologue of Russian art and culture and thus could not permit himself to remain friends with "traitors" who had joined the enemy camp—the Academy of the Arts. The renewal of relations between the artist and the critic was based on Stasov's ideological concepts. It was not until Repin made the decision to attack Stasov's ideological opponents that the latter was prepared to reestablish mutual relations. In 1899 Repin and Stasov joined forces to oppose a new common enemy—the Russian "decadents" Alexander Benua and the *Le Monde de l'art* magazine (Repin and Stasov 1950, pp. 24–29).[3] In his article "An Astounding Miracle," Stasov described how Repin rose from the dead and divested himself of incorrect opinions that had possessed him like evil spirits.

Thus one can see that Stasov's mentoring of Repin was based on his personal ideology, and this was well expressed by the nationalist and aesthetic content in Stasov's article on Repin's *Burlaki*. When the painting was exhibited for the first time, Stasov was involved in a debate with sections of the Russian intelligentsia concerning the cultural roots of the Russian people and of authentic Russian art and culture. The debate, political in nature, began in 1868 when Stasov published a study on the origins of Russian folk epics (*bilini*) (Pyzhikov 2019, pp. 26–77). These stories were considered to be the heart of the Orthodox legend of the miraculous conversion to Christianity of peoples on the borders of the Russian Empire. Stasov demonstrated that the narratives were derived from Oriental folk tales and

his research engendered a heated controversy in regard to those of his claims that challenged the fundamental beliefs of the Russian Orthodox Church and the Russian bourgeoisie.[4]

Stasov was involved in a dispute concerning the origins of the various cultures that characterized the Russian Empire for more than twenty years, his principal opponents being Slavophile ideologues, who attributed a Byzantine origin to folk tales from the imperial territories. Stasov claimed that these traditions, which the regime and the religious establishment were at pains to attribute to Slavic origins, were actually associated with the various peoples who made up the Empire. He contended that the folk tales had been rewritten to convey a monolithic picture of the Moscow-based religious establishment and to support his position he needed a realistic Russian national painter from the periphery. The young Repin fitted the bill, and his paintings reflected Stasov's message. In his magnum opus from 1883, *Twenty-Five Years of Russian Art*, Stasov argued that realist Russian art had developed along its own lines earlier than Western Realism (Stasov 1906, pp. 521–25). He declared that Chernyshevsky and Gogol were responsible for the introduction of realism in Russian art and literature and that they preceded Courbet, Millet, and the Barbizon School. In 1883 Stasov described what the literary critic Erich Auerbach calls in his classic study *Mimesis*. Auerbach characterizes the developmental difference in the perception of reality by the peoples of the Russian empire, a perception of the nation as a family, free of class and ethnic distinctions.[5]

Stasov greatly admired the technique of the painters whom he called "the French moujiks" to highlight their common origins. He called Millet "the French Repin," thus giving Russian art pride of place (Stasov 1906, pp. 548–49). Russian realism was focused on nationality, unlike the French version, which criticized the class differences. Both Courbet's *Stone Breakers* and Repin's *Burlaki* depict lower-class people hard at work, but with a significant difference: Courbet's figures are devoid of identity, their faces not clearly visible. *Burlaki* images have a clear identity, with each member of the colorful group representing an individual narrative, and collectively comprising national heroes of Russian society. The central theme of Courbet's work is a hopeless transgenerational story: the son is engaged in the same hard labor as the father, which future generations will continue in their turn. In contrast, the young boy in *Burlaki* symbolizes change and the awakening of rebellion among the lower classes. The concept of a moral critique of class discrimination was characteristic of the Peredvizhniki in the early 1860s, but it became a more nationalist movement during Alexander III's reign. Stasov called for Russian nationalist art as early as in the first years of the 1870s, and Valkenier points to him as a central figure in this change of perception (Valkenier 1975, pp. 247–65).

Stasov saw Russian nationalist art as a reflection of the national stories of the periphery and contended that *Burlaki*, was an outspoken representation of Russian heroes because the painting granted them identities. He used *Burlaki* to express his own worldview. He pointedly ignored the 'negative' allusions that the painter chose to include in his work, such as the upside-down Russian flag, symbolizing the Russian nation's distress and the hauling rope attached to the mast in such a way that hauling it would cause the barge to capsize. Stasov made no reference to the stone tied to the rope in the foreground of the work, a symbol of a floater in Russian culture. Stasov focused on the heroes, controversial but significant (who, on the one hand, are courageous and authentic and may be true comrades, but, on the other hand, might be cold-blooded murderers), on account of the "genuine article" that they represented.

Stasov constructed an equation for Russian art that was a representation of Russian heroes from the margins of society rendered in a realistic style as a sublime subject and one that should be the focus of the national painting. Thus, for Stasov, *Burlaki* with its group of seasonal workers became the model for Russian painting. As a result of Stasov's article, Repin's work became the supreme expression of Russian painting expressing the strength of the Russian people beyond the imperial capital. Stasov's ambition to promote realistic Russian national art coupled with Repin's wish to become a nationally recognized painter led to Repin using Stasov's recipe for national art.

### 3. Repin's Success as a Result of Stasov's Article

As Repin himself described in his memoirs, opinions were divided among the viewers who saw the picture in the 1873 exhibition, however, it was Stasov's article that made *Burlaki* famous and Repin a national hero. "The chief evangelist of the painting [*Burlaki*] was the noble knight Vladimir Vasilievich Stasov. His first steadfast voice was heard all over Russia, and his proclamation was heard by every Russian. [Stasov's voice] initiated my reputation throughout mighty Russia" (Repin 1944, p. 282).

Immediately upon the article's publication, Repin was transformed from a graduate of the Academy of the Arts to the creator of the ultimate nationalist artwork. Fidelio Bruni, an academy painter and rector of the Academy from 1856 to 1871, was so disturbed by the nature of Repin's work that he called it "the ultimate profanation of art," but in spite of Bruni's opinion, it was recognized by the Academy only two years later (Shumova 1995, pp. 50–54). As early as in 1875, the work became an object of imitation and study and Grand Duke Vladimir Alexandrovich frequently loaned it to the Academy so that students would have the opportunity to copy it.

In 1894, two decades after Repin became well-known, the Russian painter of his period *par excellence*, the Imperial Russian Joint Stock Company Goznak, a government body subordinate to the Ministry of Finance and responsible for issuing securities, produced an album of his works. The introduction to the album begins by describing Repin as the genius of his time whose work is the image of Russian national history (*Russie Khudogniki: Il'Ya Efimovich Repin* 1894, p. 1). Clearly, then, within twenty years, during his own lifetime. Repin had become the Russian painter of his period *par excellence*. In his 1901 survey of the history of Russian art, the art critic Alexander Benua called *Burlaki* Repin's masterwork and declared it the epitome of Russian realism (Benua 1902, pp. 175–85). In *Russian Art*, Nikolsky contended that Repin's role in Russian art was equivalent to that of Dostoyevsky in literature (Nikol'skiy 1904, p. 149). All this recognition began with Stasov's article and it was the paintings in which Repin implemented Stasov's philosophy that earned him the title of the Genius of Russian national art in his time.

### 4. A Taste of Foreign Cooking: The Failure of a Parisian Café

The success of *Burlaki* led to great expectations from Repin. As an ambitious painter, he decided to be practical in following the figurative ingredients that appear in the second part of Stasov's recipe. In fact, one can find Stasov's artistic definitions implemented in Repin's next major work, *A Parisian Café* (Figure 2), which he painted in Paris while visiting Europe on a grant from the Russian Academy of Art (Valkenier 1990, pp. 59–66). As in *Burlaki*, Repin painted a collection of characters, here a group portrait in a café in Paris. He closely followed Stasov's recipe—the setting forms a backdrop for the characters representing the daily life of the French people as a representative national group, each figure with its own story. In a letter to Tretyakov, Repin wrote that he painted quintessential Parisian characters in the quintessential Paris location (Repin's letter of 3 April 1874, to Tretyakov, in Repin 1946, p. 26). Like the figure of the youth in the center of *Burlaki*, whom Stasov singled out as a symbol of rebellion, at the center of this work Repin placed a woman defying social conventions by entering the café unescorted. Repin was excited to have understood the essence of life in Paris, just as he was able to reflect Russian life in *Burlaki*. He felt success in the air, worked unceasingly, and applied to Stasov and Tretyakov for financial assistance. He wrote to Stasov that this work would win him the reputation in Paris that he needed to sell his works there (Repin's letter of 4 March 1874, to Stasov, in Repin and Stasov 1948, pp. 88–89).

Repin was so sure that his work would be as successful as *Burlaki* that he ignored Russian Art Academy regulations by exhibiting it in a Parisian salon in 1875 (Jackson 1998, pp. 394–409). However, the work was a failure in Russian circles. The painter Arkhip Kuindzhi, who saw the painting in the salon, reported that not only was Repin's work a fiasco, but that it was highly embarrassing for the painter. His friend, the painter Ivan Kramskoi, characterized the painter's ambition to create a painting whose subject was Parisian life as an "attack of insolence" (Kramskoi's letter of 20 August 1875, to Repin, in

Kramskoi 1888, pp. 262–64). He claimed that a painter with Ukrainian blood flowing in his veins was capable of painting wild commoners (a reference to *Burlaki*), not Parisian courtesans. Kramskoi claimed that to create something genuine about Paris, one has to have grown up listening to French chansons. Kramskoi's letter continued to attack Repin. He wrote that until the appearance of *A Parisian Café* he believed that Repin had the "true string"—nationalism—and went on to point out that the proper topic for a Russian painter is one connected to Russian nationality. Although Repin defended himself and responded to Kramskoi's criticism, he took it to heart. Stasov did not publish any response, although he visited Repin in Paris and almost certainly saw the work.

Repin loved Paris all his life and continued visiting there, but he never returned to foreign subjects in his painting, except on one occasion, on a visit to Paris with Stasov on 15 May 1883, when he produced a relatively small work documenting *The Annual Memorial Meeting Near the Wall of the Communards in the Cemetery of Père-Lachaise in Paris* (Terkel 2019, n.p.). Clearly, the Russian artistic clique's reaction to *A Parisian Café* defined for Repin the ideological line that he had to follow as the painter who created an artwork like *Burlaki*: an ideology dealing with the Russian nation.

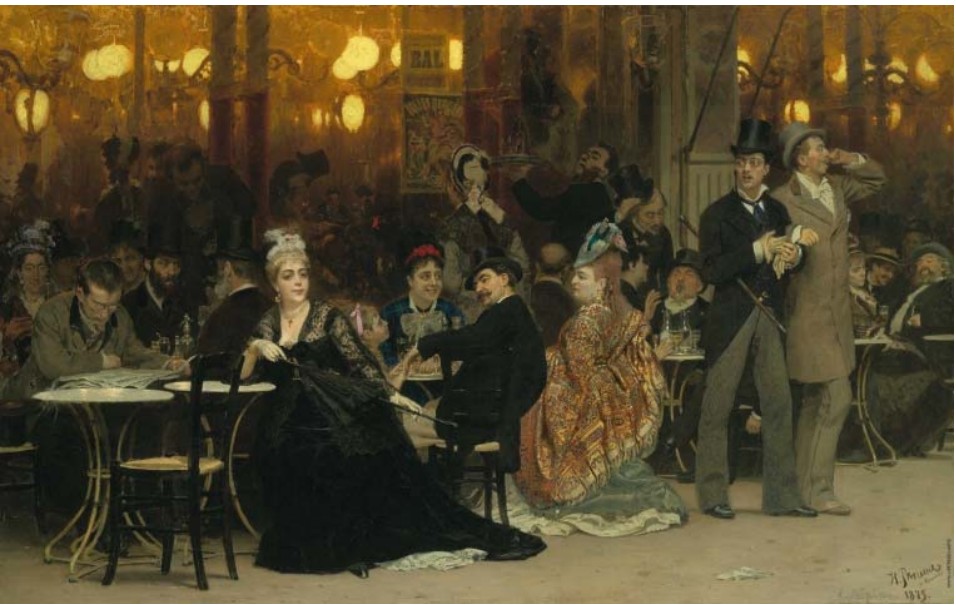

**Figure 2.** Repin, Parisian Café, 1875, Private collection, The Museum of Avant-Garde Mastery. Digital Image: Wikimedia Commons, Public Domain.

## 5. Religious Procession in Kursk Governorate: Faith Haulers

Repin returned to "cooking" his "blockbusters" according to Stasov's recipe. This is evident in his iconic works *Religious Procession in Kursk Governorate* (Figure 3) and *Reply of the Zaporozhian Cossacks to Sultan Mehmed IV of the Ottoman* Empire (1880–1891). Tretyakov paid 10,000 rubles for the former before Repin had even completed it (Grabar 1964, vol. 1, pp. 188–230). Stasov was convinced that the work would be Repin's next success and gave the painter every encouragement. The subject of the painting is a traditional religious event—the festive procession with an icon, which according to Orthodox belief can heal any illness. In the work we see a large number of figures, dozens of whom are clearly visible down to the last details of their costumes. They include men and women of all ages—bourgeois, members of the clergy, police, peasants, and poor people, who are all in motion on a country road, some of them are carrying ritual objects. The procession moves beneath a bright summer sun through clouds of dust.

One can notice the resemblance to *Burlaki* immediately—the same narrow country setting under bright sunlight acting as a backdrop to the event, and the same unified group, depicted in detail but appearing as a single unit. One can construct a narrative around each

figure in the work. Like *Burlaki*, which Stasov described as moving forward and backward, in *Procession* the forward movement of the group is perceptible. At the center of the work there is a hunchbacked child who has a similar role to that of *Burlaki*'s rebellious youth. Unlike the rest of the figures in the procession, who seem to be habitually "dragging" themselves along the route, the hunchback seems dynamic. He is rendered as one who has a genuine belief that a miracle will occur and that he will be cured. Exactly as called for in Stasov's recipe, the work deals with a national theme on the periphery: a popular festival, which might be seen as embarrassing by residents of Moscow, that unites the inhabitants of a small town who are marching through a dusty rural landscape in order to be part of a tradition that they do not particularly believe in, but are participating in as an important social occasion. The human mosaic comprises a group image of the real inhabitants of the Empire and serves as a sampler representing different strata in the society.

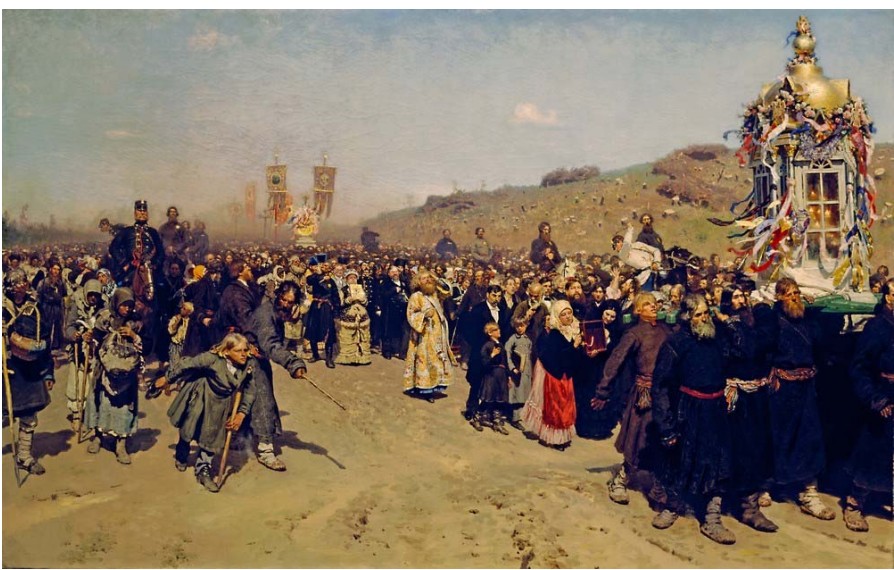

**Figure 3.** Repin, 1883, Religious Procession in the Kursk Governorate, Moscow, Tretyakov Gallery. Wikimedia Commons, Public Domain.

Repin began work on *Procession* immediately after his return to the region of his birth—Chuguev, Ukraine. The first sketch, today in the State Russian Museum in St. Petersburg, was painted in 1877 (Figure 4).

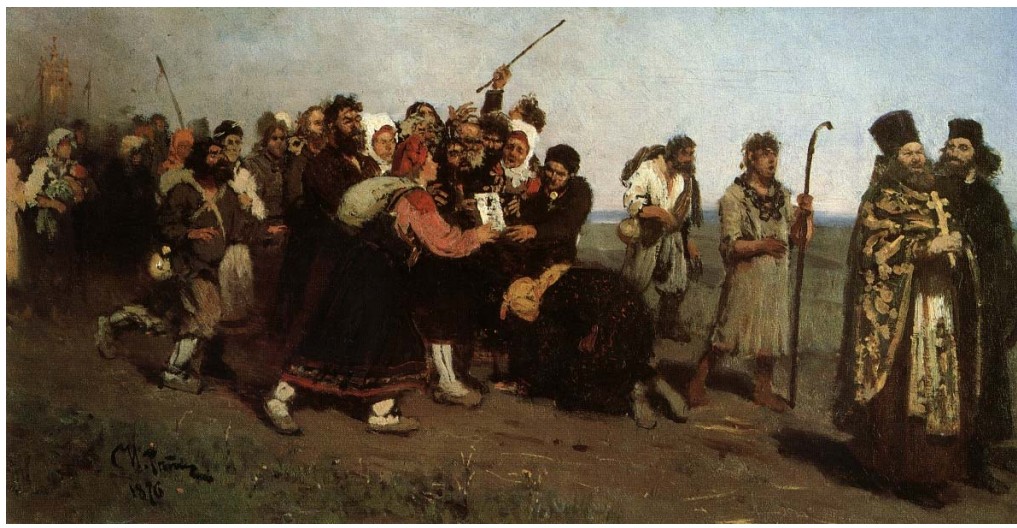

**Figure 4.** Repin, Religious Procession, 1877, Saint Petersburg, The State Russian Museum. Digital Image: Wiki Art, Public Domain.

Here, as in *Burlaki*, we see a group of figures, appearing as one elongated block within a narrow landscape. The focus is on the icon. Even though this is a preliminary sketch, it is already possible to distinguish an assembly of characters. As for *Burlaki*, Repin went to local villages to look for models to make up a scene of commoners. After the failure of *Parisian Café*, he acted cautiously, and before incorporating the characters in a large-scale work, he created portraits, exhibiting the first of them—Protodeacon (a type of clergyman) Olenov (Figure 5)—in an 1878 exhibition of the Peredvizhniki.[6]

That was the first time that Repin exhibited with this group and it received an enthusiastic reception from Stasov, who praised the work and especially the figure of the Orthodox clergyman to whom he devoted two pages (Stasov 1937, vol. 1, pp. 246–48). He described the priest as a problematic, dangerous, but friendly hero. He then responded to those whom he heard speaking against over-realistic portraiture and were disgusted by the priest's portrait, saying they would never be able to hang such a painting in their living room. In a rage, Stasov wrote: "I replied out loud to [the viewers of Repin's painting] that in that case we would have to clear out of museums hundreds of paintings by the 17th century Dutch masters, which are also 'disgusting.' What is really disgusting is the nauseatingly glamorized paintings of nymphs and Graces which the 'great aficionados of art' would be sure to hang up in their offices for evermore. That's what [the viewers] really need. All those glossy lying lumps of emptiness don't disgust them. But an honest representation of the world—that they see as intimidating and inappropriate" (Stasov 1937, vol. 1, pp. 246–47).

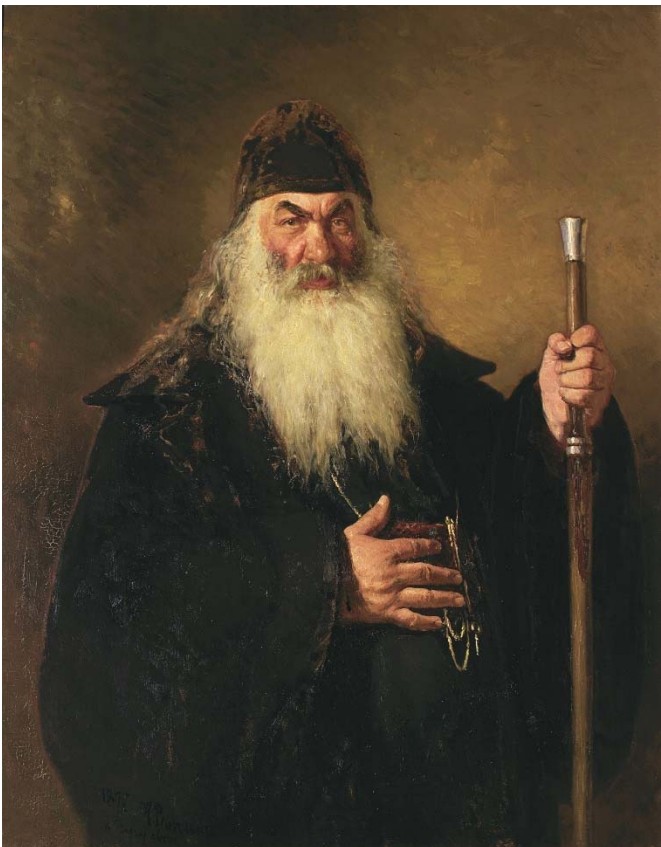

**Figure 5.** Repin, Archdeacon, 1877, Moscow, Tretyakov Gallery. Digital Image: Wiki Art, Public Domain.

Stasov's encouragement did not produce the desired effect. Instead of continuing to work on the painting, Repin decided to do other works. This led to a cooling of relations between the two. Repin exhibited a historical work, *Sophia*, which he had worked on in parallel, and received a negative review from Stasov.[7] He criticized Repin's choice of a historical theme, believing that Repin did not have the skill to handle a non-contemporary

subject. Stasov maintained that skill in historical painting is an unusual and inborn skill, and claimed that Repin's attempt to portray a historical figure was a failure because Repin was a realist and was unable to invent something he had not seen with his own eyes. Stasov took the opportunity to criticize many of the painters who had worked in this genre, including Raphael, Rubens, and Titian. "When we examine the good points of this painting [*Sophia*] . . . we want to ask when Mr. Repin will return to genuine work. When will he give us another perfect work like *Burlaki*?" wrote Stasov in response to the 1879 exhibition (Stasov 1906, vol. 1, pp. 709–12).

This negative reaction provoked Repin to invest all his energy in finishing *Procession*, which featured all the ingredients of Stasov's recipe. In *Procession* he fulfilled Stasov's requirements, and when Tretyakov (who had purchase the painting) requested Repin to "remove the ugly characters" from the foreground of the work, Repin declined the request, claiming that the scene should reflect reality.

The work was exhibited in 1883 in the Peredvizhniki exhibition and, like *Burlaki*, was the subject of an enthusiastic critique by Stasov, who wrote that the work is equal to *Burlaki* in terms of both its national power and artistic talent. He noted all the points of resemblance between the two paintings and declared that *Procession* is "the greatest celebration of contemporary art" (Stasov 1906, vol. 1, pp. 726–28).

### 6. Reply of the Zaporozhian Cossacks to Sultan Mehmed IV of the Ottoman Empire: 1880–1891—Proud National Haulers

Repin reached the peak of his maturity with *Reply of the Cossacks* (Figure 6). This 4-m wide work depicts a letter being written on a battlefield. At the center of the work, the letter writer sits at a table surrounded by more than twenty male figures, all in traditional seventeenth-century Cossack dress, some carrying various weapons. Most of the men are laughing, some apparently uncontrollably. The viewer can identify the visual elements of the Stasov-Repin partnership: a narrow landscape acting as a backdrop, a group of representative characters acting together, movement of the characters, and a central figure bearing a message of hope and rebellion. Repin showed the finished work in a solo exhibition mounted in 1891 in St. Petersburg to celebrate his career over the previous twenty years. Czar Alexander III purchased the work for 35,000 rubles, the highest sum paid up until then for a Russian painting (Grabar 1964, vol. 2, pp. 80–82). The art critics Stasov, Benua, and Dobzhinski praised the work in the press, and it was immediately hailed as a great success (Jackson 2015, pp. 91–95).

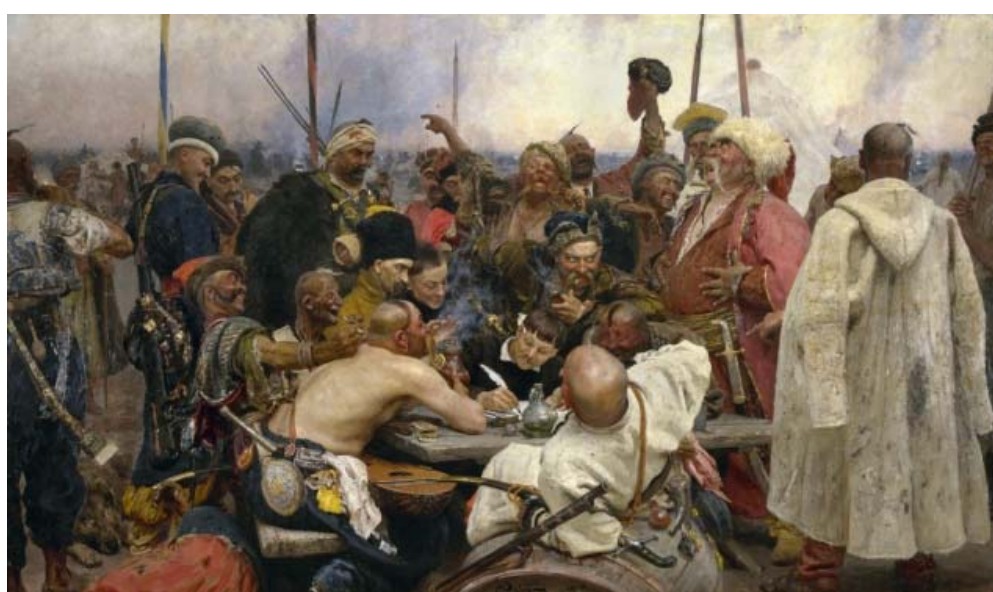

**Figure 6.** Repin, Reply of the Zaporozhian Cossacks to Sultan Mehmed IV of the Ottoman Empire, 1891, Saint Petersburg, State Russian Museum. Digital Image: Wikimedia Commons, Public Domain.

Repin's choice of subject and his method of choosing models were entirely in line with the nationalist ideology that Stasov strove to uphold. As when working on Burlaki and Procession, Repin employed "actors" for the work. In Stasov's view, an actor or model had to match the spirit of the character not only in visual terms but also in his/her essential nature. In Russian culture, the Cossacks epitomized authenticity and free-spiritedness not subservience to the Empire: lords of creation, so to speak.[8] Repin chose them because they resembled the figures in Burlaki in many respects. In present-day scholarship concerning Repin's oeuvre, this work is assigned to the historical genre. The work is based on a legend according to which during the war between Russia and Turkey in 1672–1681, the Cossacks responded to a call for surrender by the Ottoman Sultan by composing a mocking letter filled with curses.[9]

David Jackson writes that Repin chose the Cossacks because they were close to his heart, since his birthplace was in Zaporozhye, the region where the Cossacks originated. The idea of painting Cossacks came to him after he saw a letter by the Cossack leader Ivan Sirko but there were apparently additional reasons for the choice of subject beyond an Orientalist depiction of the painter's birthplace. The Cossacks of Zaporozhye featured in Gogol's famous 1842 novella *Taras Bulba*, and that protagonist is featured on the right-hand side of Repin's painting. As will be remembered, Stasov had opened his article on *Burlaki* with a comparison between Repin and Gogol, so the artist decided to respond to Stasov's comments by choosing a subject from the work of an author whom Stasov admired. A letter by Stasov addressed to Repin dated 23 November 1888, shows the extent to which Stasov was following Repin's progress on the work. He not only encouraged him but also suggested unifying the figures of the Cossacks around a significant diplomatic action that they were treating freely and comically (Repin and Stasov 1949, vol. 2, p. 136). In a letter to Repin on his birthday dated 15 July 1890, Stasov greeted the artist by mentioning the painting and encouraged him to complete it. He compared it to the seventeenth-century Spanish painter Diego Velázquez's *Las Lanzas* in that the figures are similar to Repin's Cossacks on the battlefield. Repin seems to have had trouble creating images of "realistic national heroes" as for *Burlaki* because his subjects were historic rather than contemporary. But he kept to his familiar recipe for success and achieved his goals by a clever use of models from a similar background to the figures in the painting. He based himself on his own canonical example in *Burlaki* and constructed the work accordingly.

As a model for the central figure, Cossack commander Ivan Sirko, Repin selected Mikhail Dragomirov, a Russian general who was one of the outstanding strategists of the second half of the nineteenth century and played an active role in the 1877–1878 war between Russia and Turkey. For the leading role of the scribe at the center of the work he employed Dmytro Yavornytsky, a historian, an expert on Ukrainian ethnology and scholar concerned with Cossack heritage. It is interesting that Repin chose a well-known nineteenth-century historian to embody the role. He represents the only literate character in the picture and is depicted writing a historic document—a position paper from the Russian people directed against foreign nations. There may be an allusion here to Gogol himself, who worked as a clerk. This figure plays the same role in the work as the boy in *Burlaki*: just as the boy there represents the younger generation's rebellion and the hope of new Russia, the scribe symbolizes the fact that the new Russian history, unafraid of the powerful enemy, is written by the Russian intelligentsia.

Another figure that attracts the viewer's attention is a tall Cossack with a bandaged head—the wounded Cossack's reaction to the writing of the letter is particularly impassioned. According to Gogol's story, this is Taras Bulba's son, who is captured and executed in front of his father at the end of the novella. The model for this figure was Nikolay Kuznetsov (1850–1929), an academist painter.

The decision to take models from the Russian intelligentsia and portray them as Cossacks representing the epitome of heroism and power from the seventeenth century was intended to glorify Russian nationalism, using models from Repin's own day, signifies that the heroes of the work are members of the Russian intelligentsia. As noted, Repin

revived the story of the legendary Cossacks on the basis of Gogol's novella. Here, too, he was following Stasov's nationalist vision and combining figures from Russia's heroic past with contemporary cultural heroes—a Russian general, an academist painter, and a historian. With the creation of a group portrait of his own contemporaries dressed as Cossacks, Repin used the Cossack national epic to convert a historical painting into a portrait of contemporary Russia.

## 7. Conclusions

The publication of Stasov's 1873 article in the correspondence column of the daily newspaper *Sankt-Peterburgskie Vedomosti* turned Repin's Burlaki *na Volge* into a canonical work. By reimplementing Stasov's recipe, Repin achieved fame as a national artist in his own lifetime and, in fact, became the ultimate Russian painter. In his first acknowledgment of the painter, Stasov achieved two objectives: he turned Repin into Russian national epitome and enlisted the painter's talents to espouse the concept of Russian national art that he wished to promote. One may conclude that Repin's success was the result of his implementation of Stasov's recipe. In several of his works, the painter chose to employ only some of the ingredients, but whenever he was determined to fill the position of Russian national painter that Stasov had proclaimed him to be, he went back to the list of Stasov's ingredients and as the works appeared he achieved fame as the chef who satisfied the critics' palate.

**Funding:** This research received no external funding.

**Institutional Review Board Statement:** Not applicable.

**Informed Consent Statement:** Not applicable.

**Data Availability Statement:** Not applicable.

**Acknowledgments:** I would like to thank to Daniel M. Unger for his kind guidance and priceless advice. I am grateful to Nirit Ben-Aryeh Debby for wonderful ideas, and to Simon Montagu for his help with English translation. I am thankful to my reviewers for intellectual reading and valuable insights.

**Conflicts of Interest:** The authors declare no conflict of interest.

## Notes

[1]　For example, see Stasov articles about the Annual Academic Exibitions (Stasov 1937, pp. 18–28)

[2]　The Russian philosopher Karasev explains the place of the stomach in Gogol's writings as the center of the universe (Karasev 2012, n.p.).

[3]　*Le Monde de l'art*, published in St. Petersburg from 1898–1904, was a monthly journal devoted to publicizing modern genres in Russian art. Stasov and the journal's editors, Diaghilev and Benua, had frequent clashes on issues relating to the nature of art in general and of Russian art in particular. See further Melnik (2015).

[4]　For more on Stasov's revolutionary studies, see the work of the Russian historian Alexsander Pyzhikov (Pyzhikov 2019).

[5]　Auerbach's study wherein he characterizes the developmental difference in the perception of reality by the peoples of the Russian Empire is referenced below (Auerbach et al. 2013).

[6]　About Repin and Peredvizhniki in the years 1870–1890, see (Jackson 1990, pp. 8–22) and (Zograf 1977, pp. 69–82).

[7]　Repin, *Tsarevna Sophia Alexeevna in the Novodevitchy Convent*, 1879, Moskow, Tretykov Gallery.

[8]　An interesting question is why the conservative Russian Emperor Alexander III chose to purchase a work centered on a force of Cossacks disobeying the laws of the Russian Empire. For an answer, see the beginning of the article by art historian Walther Lang, "The Legendary Cossacks: Anarchy and Nationalism in the Conceptions of Ilya Repin and Nikolai Gogol," where he notes that the reason was apparently the patriotic topic and the charismatic characters (Lang 2002, n.p.).

[9]　The Cossacks' letter was full of the most picturesque curses imaginable and extremely humiliating expressions directed at the sultan. For a reference to the original text in Ukrainian and in Russian: https://www.rugrad.eu/columns/Kazaki/8578/ (accessed on 11 November 2021) and for an English translation by the historian Andrew Gregorovich: https://unfcanada.ca/wp-content/uploads/2020/06/MYH-BEAMS-v4n1-Jan.-1958.pdf (accessed on 11 November 2021).

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
