# Peer review of "The Recipe for Success: Repin’s Painting Barge-Haulers on the Volga and Stasov’s Conception of Russian Art"

_arts_

Round 1
Reviewer 1 Report
The article is coherently put and balanced. It represents a welcome addition to the scholarship on Repin.
Some concrete suggestions that are aiming to become additions to the text:
1) To add some four-three passages of discussion on the way Russian realism or "critical realism" different from the Western/European one. What is the difference between Repin and Delacroix? A good comparative addition is definitely needed as otherwise it is impossible to understand the realism of Russian Peredvizhniki. This must include also at least a little some theoretical consideration on what Realism is and what place Mimesis occupies in it. Auerbach's classical study must be widdely quoted and cited.
2) When discussing the letter of the Kazaks to the Sultan a full text of it in translation must be appended since Repin carefully wrote all the content of it in the picture. A full translation of this short letter is the ideal addition, it is not too big. At any case a very detailed retelling of the content must appear with all the obscene words to be published without any taboo or omissions since this is academic scholarship.
The text of the letter is universally available: https://www.rugrad.eu/columns/Kazaki/8578/
Reviewer 2 Report
Very intriguing and original analysis of Stasov and Repin.
The following comments are meant to be helpful and go in order of the article.
The first sentence in the abstract is missing a verb, perhaps use IS "....on Volga IS one of....."
Perhaps use "hampered" rather than "constricted." [line 6]
What is meant by "a typical group of paintings?" [line 13]
Is the actual title "Burlaki" or Бурлаки на волге?"
"shown" not "showed" [ line 22]
Perhaps not "a simple day to day task" but "peasants working" [line 23]
What was the title of the competition? [line 26]
What is the 1873 exhibition? [line 32] Is it the competition [line 26]
What is the title of Stasov's article in SP Vedomostii? [31]
What is mean by "venue" [33]
What is the date of Grabar's monograph on Repin [42]
What is meant by "mood swings" [56-57]
What are the parameters for his "successful" works [72]
Be more specific about who the "boy" is and, perhaps, introduce this idea earlier in the paragraph [88]
Perhaps include a quote from Repin when he describes Repin's work "at great length' [94-95]
Why did Stasov have an "ideological conflict" with the Academy? Perhaps you can explain this in one sentence. [98] (This would also help when you describe the Academy as his bete noire later in the paper)
In what way are the figures typical of Russian society? [108]
Not quite clear what is mean by "tone and temperature." [109]
Perhaps one sentence explanation on how a "narrow landscape" distances national art from academic art traditions. [121-122]
Not clear to what pyramid you are referring. [137]
Not quite clear is it Stasov challenging in the "pages of the newspaper (which?) and in letters" or is it the other artists and creators. If the latter, specify who. [138-139]
Not quite clear about the "renewal....based on Stasov's ideological concepts" [155-156]
What was Stasov's argument about the origins of the folk epics? [166]
Did Stasov "prove" the studies connect to various peoples of empire? Prove is a strong word on something that is still an issue of debate. [footnote number 3 at bottom of page 4] Also, may wish to put this footnote section in the body of the paper.
Quite a bit of French Realism did show the common person so may be a stretch to say this was not the case [178]
Is the stone on the rope a symbol of suicide or murder in Russian culture? If so, state this explicitly. [186]
Not quite clear how "heroes" are "controversial but significant" for Stasov. [187]
How was Repin's work "recognized by the Academy?" [208]
On what "mission" was Repin [234]
Not quite clear why Repin thought doing a work of Paris, despite adhering to the 'recipe' laid out by Stasov, would suffice.
Not clear what is meant by "infringed" [248]
What was the date and work of the "one occasion" Repin painted foreign subjects [264]
Be consistent with title of Religious Procession, has two different titles in paper [see 271 and 273]
Why in Religious Procession is the "national theme on the periphery" rather than in the center as in Burlaki [292]
Where was the region of the his birth? [296]. It is mentioned later in the paper but might be better put here.
What is meant by "went fishing for peripheral characters?" 305
In English the standard transliteration is "Peredivizhniki" not "Peredvigniki", which shows up in several places [309 and footnote in 4, 330]
What was Stasov's response? [314-316]
Why did Stasov criticize Repin and historical themes? Was it only he did not think he had the skill to do it? ]321-322]
Did Tretyakov really commission the work? [325] It seems earlier in the article he bought it before he even completed it. [273]
What was Repin's "method of choosing models?" [353]
If the work (Zaporozhian Cossacks) showed that the Cossacks were not subservient to the empire why would the Czar purchase the work? Why did he purchase it?
Perhaps state where Repin found the letter by Ivan Sirko and what the content of the letter was. [366]
Something seems to be missing "...figures look alike [space] Repin seems..." Not clear which figures look alike and the following section [379]
Perhaps there is no need to put the Russian transliteration of the word for "scribe" as no other words have a transliteration [387]
How does the scribe play the same role as the youth in Burlaki? [390]
How is the hero of the work the "Russian intelligentsia?" Perhaps explain in one sentence. [399]
How is the work of a Cossack national epic converting "a historical painting into a portrait of contemporary Russia?" [406]
In Conclusion, once again put the title of Stasov's article. [408]
Again, it is a good article, interesting, and original. The comments are merely to further enhance the article.
